# Surrogate Signals from Format and Length: Reinforcement Learning for Solving Mathematical Problems without Ground Truth Answers

## Abstract

Large Language Models (LLMs) have achieved remarkable success in natural language processing tasks, with Reinforcement Learning (RL) playing a key role in adapting them to specific applications. In mathematical problem solving, however, the reliance on ground truth answers poses significant challenges due to their high collection cost and limited availability. This work explores the use of simple surrogate signals, format and length, to guide RL training. We find that early training is dominated by format learning, where structural feedback alone accounts for most performance gains. Incorporating length-based rewards further refines outputs by discouraging overly long or short responses, enabling a GRPO approach with format-length signals to match, and in some cases surpass, ground-truth-based optimization. For example, our method achieves 40.0% accuracy on AIME2024 with a 7B base model, and generalizes across different model sizes and series. Beyond practical efficiency, these findings provide an inspirational perspective on RL: rather than imparting new knowledge, RL primarily activates reasoning capabilities already embedded in pre-trained models. This insight suggests that lightweight, label-efficient strategies can complement pre-training to unlock LLMs' latent potential in reasoning-intensive tasks.

## 1 Introduction

In the dynamic landscape of artificial intelligence, Large Language Models (LLMs) Brown et al. (2020); Chowdhery et al. (2023); Yang et al. (2023); Wang et al. (2025a); Grattafiori et al. (2024) have emerged as a transformative force, with models like GPT-o1 Jaech et al. (2024), DeepSeek-R1 DeepSeek-AI et al. (2025), and Qwen3 Yang et al. (2025) leading the charge. Pre-trained on massive text corpora, these models have demonstrated remarkable proficiency across diverse natural language processing tasks—ranging from text generation and question answering to translation and code synthesis. Their success largely stems from unsupervised pre-training, which enables them to capture complex semantic and syntactic structures and generalize effectively across scenarios.

Reinforcement Learning (RL) has become a pivotal technique for adapting these pre-trained LLMs to specific downstream tasks. Popular algorithms such as Proximal Policy Optimization (PPO) Schulman et al. (2017) and its advanced variant, Group Relative Policy Optimization (GRPO) Shao et al. (2024), are widely adopted to refine model behavior. Conventionally, these methods rely on ground truth answers as rewards, providing explicit supervision for iterative optimization. Yet in domains like mathematical problem solving, obtaining ground truth is costly, labor-intensive, and often infeasible due to the scarcity of accurate annotations. This bottleneck has motivated the search for label-free reinforcement learning frameworks that can effectively improve reasoning ability without requiring explicit ground truth.

Our work is inspired by a key empirical observation: the early stages of RL training are dominated by **format learning**. Within the first ~15 optimization steps, models rapidly converge toward structured and concise solutions, yielding over 85% of the total performance improvements while drastically reducing redundancy. Strikingly, rewards based solely on format correctness already

achieve comparable gains to standard GRPO with ground truth, underscoring the surprising potency of structural feedback. However, such **format-only** optimization soon saturates, as it fails to guide the model beyond structural compliance.

To address this limitation, we introduce a **Format-Length** reward that augments format correctness with constraints on response length, penalizing outputs that are excessively long or short. Together, these surrogate signals are strongly correlated with correctness, enabling performance that not only matches but sometimes surpasses GRPO trained with ground truth. This highlights the potential of surrogate rewards as lightweight yet effective substitutes for explicit supervision.

Our perspective is grounded in the assumption that modern base models already possess substantial mathematical competence. For instance, Qwen2.5-Math achieves high pass@64 accuracy but underperforms under strict pass@1 evaluation, akin to a well-prepared student whose knowledge is underutilized due to inefficient problem-solving habits. In this analogy, format-length rewards act as training strategies that "activate" latent capabilities rather than instill new knowledge.

This leads to a central question: *Do reinforcement learning gains primarily arise from acquiring new knowledge, or from surfacing knowledge already embedded in the base model?* Our findings support the latter. The implications are noteworthy: pre-training should be viewed as the primary stage for knowledge acquisition, while RL serves as a lightweight mechanism to unlock this latent knowledge for downstream reasoning tasks. By showing that surrogate signals can replace explicit ground truth in mathematical problem solving, our work provides an *inspirational perspective* on how LLMs can be efficiently post-trained in scenarios where ground truth answers are scarce, highlighting a promising direction for future research in label-free reinforcement learning.

## 2 METHOD: FORMAT AND LENGTH AS SURROGATE SIGNALS FOR ANSWER

To mitigate the issue of label scarcity in real-world environments, we explore the potential of format and length as powerful "surrogate signals" highly correlated with answer correctness. Format correctness in mathematical problem-solving offers a necessary but insufficient condition for answer accuracy, providing a clear structural optimization target for the model. Meanwhile, the length of the response serves as an indicator of content efficiency and logical compactness, reflecting the quality of the solution's reasoning process. Based on these insights, we develop a novel learning framework that integrates format and length rewards into the GRPO algorithm. This framework, centered around optimizing LLMs without relying on explicit ground truth answers, aims to enable effective training by leveraging these surrogate signals to approximate the optimization direction of ground truth answer rewards.

### 2.1 FORMAT REWARD

In the context of mathematical problem-solving, a correct format is crucial for ensuring the clarity and comprehensibility of the solution. Our format reward mechanism is designed to encourage the model to generate responses that adhere to the standard presentation conventions of mathematical solutions (details in Appendix A.8). The format reward $R_{\mathrm{f}}$ is defined as a binary function:

$$R_{\mathrm{f}} = \begin{cases} 1 & \text{if the format is right.} \\ 0 & \text{else.} \end{cases} \tag{1}$$

This reward serves as a fundamental signal for the model to learn the structural aspects of mathematical problem-solving in the early stages of training.

### 2.2 LENGTH REWARD

To complement the format reward and further refine the content of the model's responses, we introduce a length reward function. The length of a response is a critical factor that reflects the efficiency and logical compactness of the solution. An overly short response may lack essential reasoning steps, while an excessively long response might contain redundant or incorrect derivations.

Our length reward function is designed to strike a balance between promoting comprehensive reasoning and preventing overly long responses that could exceed the model's context limits. It is

formulated as a piecewise function:

$$R_{\mathrm{l}} = \begin{cases} 1 - \left(1 - \frac{x}{p}\right)^2, & 0 \le x \le p, \\ 1 - 2\left(\frac{x-p}{1-p}\right)^2, & p < x \le 1, \end{cases} \tag{2}$$

Let

$$x = \frac{L}{L_{\max}}, \tag{3}$$

where $L$ is the length of the current response and $L_{\max}$ is the maximum context length. Let $p \in (0, 1)$ be a tunable parameter that controls the turning point of the piecewise function, with a default value of 0.5. This piecewise function is continuous and differentiable at $x = p$, encouraging response lengths that approach the turning point $p$. The reward increases smoothly as $x$ grows from 0 to $p$, reaches a maximum at $x = p$, and then decreases for $x > p$, thereby penalizing overly long responses.

In order to eliminate the influence of randomness and verify the design principle of length reward "first-rise-then-drop" curve (encouraging moderate-length reasoning chains and discouraging overly short or long outputs), we designed alternative shapes such as the polyline length reward: (where x and p are defined the same as above)

$$R_{\mathrm{l\text{-}polyline}} = \begin{cases} 2x, & 0 \le x \le p, \\ 3 - 4x, & p \le x \le 1. \end{cases} \tag{4}$$

A positive length reward can only be obtained when the format is right. Examples with format errors are considered severe—no matter how ideal their length may be, they can receive at most 0. Therefore, the final format-length reward can be expressed as:

$$R_{\mathrm{fl}} = \begin{cases} R_{\mathrm{f}} + R_{\mathrm{l}} & \text{if the format is right.} \\ \min(0, R_{\mathrm{f}} + R_{\mathrm{l}}) & \text{else.} \end{cases} \tag{5}$$

By combining the format reward and length reward, we provides an "surrogate signals" for the model's reinforcement learning, helping to alleviate the issue of label scarcity in real-world environments. There is more discussion of the design of length reward in Section 4.3.

## 3 EXPERIMENTS

### 3.1 EXPERIMENTAL SETUP

**Reward configurations:** We designed a series of experiments with distinct reward configurations to assess the effectiveness of our proposed approach. **Correctness:** This configuration is served as our baseline, which uses the exact match with ground-truth answers as the reward criterion. When the model's output precisely aligns with the correct answer, it is assigned a reward score of 1; otherwise, it receives 0. We utilized the MARIO_EVAL [1] library to accurately extract answer content from the model's output, ensuring a reliable evaluation standard. **Format-Only:** The reward function is as shown in Eq.equation 1, which is determined solely by the format of the model's output. After normalizing the content, we employ SymPy [2], a powerful symbolic mathematics library, to validate its mathematical format. **Format-Length:** The reward function is as shown in Eq.equation 5, where the default length function is Eq.equation 2 and the format reward is the same as that of Format-Only RL.

**Datasets:** We trained models on two mathematical reasoning datasets: **DeepScaleR** Luo et al. (2025) and **MATH-train**. DeepScaleR (17,000 samples) integrates problems from the MATH Hendrycks et al. (2021), AMC (2023), AIME (1984-2023), and others, with deduplication and decontamination applied. MATH-train (7,500 samples) is the MATH dataset's training split.

**Evaluation:** We evaluated the model on three datasets: MATH500, AIME2024, and AMC2023 with greedy decoding. In addition to analyzing each dataset individually, we also calculated the average scores across all benchmarks to enable direct comparison.

---

[1] https://github.com/MARIO-Math-Reasoning/MARIO_EVAL
[2] https://github.com/sympy/sympy

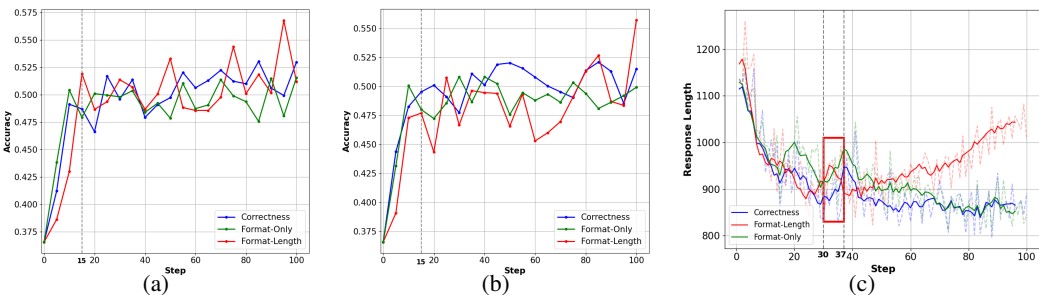

Figure 1: (a)(b) Average accuracy on evaluation benchmark training on (a) DeepScaleR and (b) Math-train. (c) Response length during training. The solid lines in the figure represent the original results, while the dashed lines represent the results smoothed with a window size of 5.

**Implementation details:** We trained the Qwen2.5-Math series base model and Llama3.1 mid-training model (OctoThinker-8B-Hybrid-Base) Wang et al. (2025c) using the GRPO algorithm under the verl [3] framework. For each case in training and evaluation, we used Qwen-Math template (as shown in Appendix A.7). During training, we used the following hyperparameters: a learning rate of 1e-6, a batch size of 128, a temperature of 0.6, 8 responses per prompt, a maximum response length of 3072, and a KL coefficient of 0.001.

### 3.2 IMPACT OF FORMAT REWARD

The format-only experiment offers critical insights into the role of format correctness in the training process. During the initial 15 steps, as depicted in Figure 1, the performance of the model trained with format-only reward remarkably aligns with that of the correctness reward setup on both benchmarks. This convergence validates our hypothesis that in the early stages of GRPO, the model predominantly focuses on learning the structural patterns of mathematical solutions. It suggests that format serves as a strong initial signal, allowing the model to quickly grasp the essential presentation conventions of mathematical answers, which accounts for approximately 85% of the overall performance improvement in this early phase.

However, as the training progresses beyond the 15-step mark, a significant divergence emerges. The performance of the format-only model plateaus, barely showing any improvement even after 100 training steps. This stagnation can be attributed to the inherent limitation of relying solely on format as a reward signal. While format correctness is a necessary condition for answer accuracy, it is not sufficient. Without additional guidance, the model lacks the means to refine the content within the correct format, leading to an inability to further enhance the accuracy of its solutions. This highlights the need for supplementary signals to drive continuous improvement.

### 3.3 EFFECTIVENESS OF FORMAT-LENGTH RL

Our format-length reward demonstrates notable advantages in mathematical problem-solving without ground truth answers, as shown in Table 1. By using format consistency and response length as surrogate signals, the approach achieves competitive performance against the model trained with correctness reward.

Numerically, Qwen2.5-Math-7B model trained with format-length reward achieves an average score of 56.8, surpassing the correctness reward's average score of 53.0 when using the DeepScaleR training dataset. In particular, model trained with format-length reward achieved 40 points in AIME2024 using the MATH training dataset.This indicates that leveraging structural and length-based rewards alone can guide the model to generate high-quality solutions comparable to or better than models trained with correctness reward, even without explicit answer supervision.

We also evaluated our method on Qwen2.5-Math-1.5B/72B, as shown in Table 2. The experiments confirm that the method is effective on models both smaller than 7B and larger than 7B. Especially, our method works on larger and more mathematically powerful LLMs (72B) can prove that the

---

[3]https://github.com/volcengine/verl

Table 1: Accuracy comparison of different models on benchmark datasets (cyan rows denote our trained models). Results are separated by a slash for DeepscaleR and MATH-train datasets (DeepscaleR first, MATH-train second). Results without * are evaluated in our environment (details in Appendix A.6); * indicates results from Liu et al. (2025b) or the original paper.

| Method | Label Free | AIME2024 | MATH500 | AMC2023 | AVG. |
|---|---|---|---|---|---|
| Qwen-Math-7B | – | 16.7 | 50.8 | 42.2 | 36.6 |
| DeepSeek-R1-Distill-7B@3k | ✗ | 10.0* | 60.1* | 26.2* | 32.1* |
| DeepSeek-R1-Distill-7B@8k | ✗ | 33.3* | 88.1* | 68.4* | 63.3* |
| Qwen2.5-Math-7B-Instruct | ✗ | 16.7 | 83.2 | 55.4 | 51.8 |
| LIMR-7B Li et al. (2025b) | ✗ | 23.3 (32.5*) | 74.8 (78.0*) | 60.2 (63.8*) | 52.8 (58.1*) |
| SimpleRL-Zero-7B Zeng et al. (2025) | ✗ | 26.7 (40.0*) | 75.4 (80.2*) | 57.8 (70.0*) | 53.3 (63.4*) |
| Oat-Zero-7B Liu et al. (2025b) | ✗ | 40.0 (43.3*) | 78.2 (80.0*) | 61.5 (62.7*) | 60.0 (62.0*) |
| Correctness (baseline) | ✗ | 26.7 / 26.7 | 74.6 / **73.0** | 57.8 / **56.6** | 53.0 / 52.1 |
| Format-Only | ✓ | 26.7 / 26.7 | 72.6 / 72.8 | 55.4 / 53.0 | 51.6 / 50.8 |
| Format-Length(polyline) | ✓ | 26.7 / - | 72.2 / - | 54.2 / - | 51.0 / - |
| Format-Length | ✓ | **33.3 / 40.0** | **76.8 / 73.0** | **60.2** / 54.2 | **56.8 / 55.7** |

Table 2: Comparison of models with different sizes and series on various benchmarks after training on the DeepscaleR dataset.

| Method | Label Free | AIME2024 | MATH500 | AMC2023 | AVG. |
|---|---|---|---|---|---|
| Qwen-Math-1.5B | – | 20.0 | 32.4 | 28.9 | 27.1 |
| Correctness (baseline) | ✗ | **16.7** | **66.8** | 45.8 | 43.1 |
| Format-Only | ✓ | **16.7** | 63.0 | 43.4 | 41.0 |
| Format-Length | ✓ | **16.7** | 64.4 | **49.4** | **43.5** |
| Qwen-Math-72B | – | 33.3 | 76.2 | 59.0 | 56.2 |
| Correctness (baseline) | ✗ | **46.7** | 80.6 | 66.3 | **64.5** |
| Format-Only | ✓ | 40.0 | 80.4 | **67.5** | 62.6 |
| Format-Length | ✓ | **46.7** | **81.2** | 60.2 | 62.7 |
| Llama3.1-8B (OctoThinker-8B-Hybrid) | – | 3.3 | 31.0 | 21.7 | 18.7 |
| Correctness (baseline) | ✗ | **16.7** | 64.6 | **34.9** | **38.7** |
| Format-Only | ✓ | 3.3 | 59.0 | 28.9 | 30.4 |
| Format-Length | ✓ | 10.0 | **64.6** | 32.5 | 35.7 |

method has some generalization and robustness. The effect of our method on LLAMA3.1 is weaker than that of the Qwen-Math series models, because format-length training requires strong inherent reasoning capabilities in the base model to be effectively activated via non-ground-truth signals, despite this, the **Format-Length** achieved a score of 92.2% of the baseline.

Figure 1 shows the average accuracy curves of GRPO training on Qwen-Math-7B with different rewards. In Appendix Figures S2 and S3, we present the accuracy curves of each benchmark respectively. It can be seen from these figures that model trained with format-length reward maintains stable performance comparable to the correctness reward baseline throughout the entire training process. The consistent curves validate the reliability of surrogate signals in driving model improvement without ground truth, highlighting the approach's scalability and data efficiency for mathematical reasoning tasks.

It is worth noting that the polyline length reward corresponding to Eq.equation 4 also achieved results close to the baseline, which shows that our method is not sensitive to the exact analytical form of the length reward, but rather to its general inductive bias.

### 3.3.1 UNCONTAMINATED EVALUATION SETS

We noticed that recent papers have raised concerns about data contamination in the Qwen2.5 series of models. Therefore, we additionally tested our method on evaluation sets released in 2025 (AIME2025, LiveMathBench). Since the Qwen2.5 series was released in 2024, it is unlikely that these datasets were contaminated. In addition, we also evaluated on the MinervaMath dataset. The three datasets mentioned above are uncontaminated, as demonstrated by the paper Wu et al. (2025). As shown in

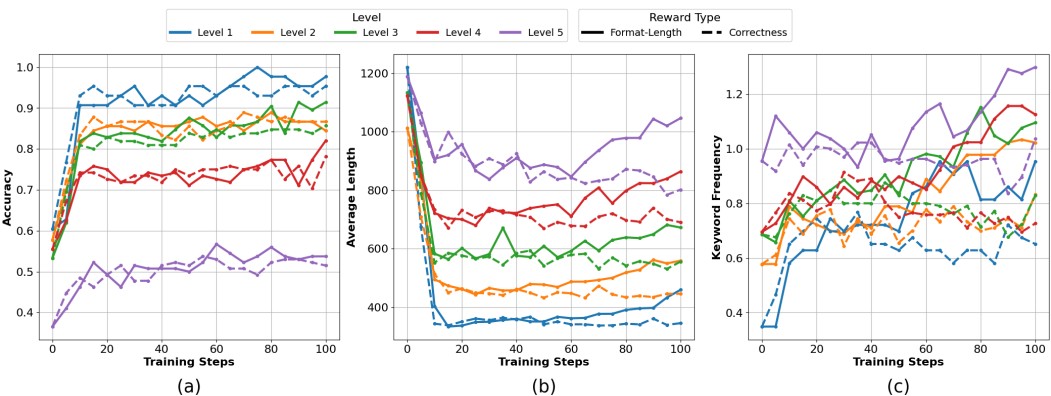

Figure 2: The curves of (a) accuracy, (b) response length, and (c) reflective keyword frequency for cases of different difficulty levels in MATH500 during training.

Table 3, our method also achieves considerable gains on uncontaminated datasets, which demonstrates its effectiveness and credibility.

## 3.4 RESPONSE LENGTH DYNAMICS

In Figures 1c, we respectively show the curves of average response length during GRPO training with different rewards on the DeepscaleR dataset. The model trained with format-length reward demonstrated a distinctive dual-phase evolution in response length, which starkly contrasts with the monotonic decrease observed in the models trained with correctness reward and format-only reward.

Across all reward configurations, the average response length decreases during the initial 30 training steps. This indicates that the model prioritizes format adherence during this phase. Driven by the dominant format reward signal, which penalizes any deviation from the required answer schema, it prunes redundant content to meet structural constraints.

As training advances from 30 to 100 steps, the length reward mechanism takes the lead, driving a strategic expansion of response content. Unlike simplistic length penalties that encourage brevity at the cost of depth, GRPO with format-length reward fosters an optimal equilibrium. It encourages longer thinking processes and discourages unnecessary verbosity. This dynamic mirrors the human problem-solving process, where initial efforts focus on establishing structure, followed by iterative refinement of content. During the final stages, the model's response length increases by an average of 14.0%, which correlates with a 10.5% improvement in average accuracy training on DeepScaleR, indicating that length serves as a proxy for reasoning complexity rather than redundancy. This dual-phase evolution parallels the human learning process encapsulated by the adage "Reading thin before reading thick." In the first stage, the model, similar to human summarization, compresses a single reasoning process, while in the second stage, it expands and generalizes, exploring more diverse and complex reasoning paths, such as error correction and branch exploration. In contrast, the correctness reward baseline and format-only models, as highlighted by the red box in Figure 1c, briefly attempt to explore complex reasoning but ultimately revert to the "comfort zone" of compressing a single reasoning process.

## 3.5 FORMAT-LENGTH REWARDS' IMPACT ACROSS DIFFICULTY LEVELS

To explore how format-length rewards affect LLMs' mathematical problem-solving, we analyzed the MATH500 dataset, which has official difficulty ratings and balanced problem distribution. As depicted in Figure 2(a), by the end of the training process, the format-length model outperformed the correctness reward baseline across all difficulty levels.

The relationship between response length and reasoning performance further illuminates the mechanism behind these results. As shown in Figure 2(b), both models generated longer responses for higher-difficulty problems. The correctness reward baseline model initially showed a rapid decrease in output length, which later stabilized, while the format-length model demonstrated a mid-stage

Table 3: Accuracy on uncontaminated benchmark datasets. The train dataset is DeepScaleR.

| Method | AIME2025 | MinervaMath | LiveMathBench | AVG. |
|---|---|---|---|---|
| Qwen-Math-7B | 3.3 | 7.4 | 5.0 | 5.2 |
| Correctness | 16.7 | 17.3 | **15.0** | 16.3 |
| Format-Only | 16.7 | 16.2 | 11.0 | 14.6 |
| Format-Length | **23.3** | **23.2** | 10.0 | **18.8** |

increase, especially for high-difficulty problems. This increase in length was positively correlated with improved accuracy, indicating that the length reward encourages the model to adopt more comprehensive reasoning strategies, particularly when tackling complex tasks.

We delved deeper into the model's reasoning process by analyzing the frequency of reflective words in the generated responses (Figure 2(c)). Reflective words, including those related to verification (`wait/verify/check`), retrospection (`recall/recheck`), branch exploration (`alternatively`), logical turn or contrast (`however/but/since`), and problem decomposition and step-by-step reasoning (`step/step-by-step`), represent complex reasoning behaviors. The correctness reward baseline model showed an initial increase in reflective words, which plateaued in the later stages, aligning with its limited performance gains. In contrast, the format-length model exhibited a significant rise in reflective words, especially for high-difficulty problems. This indicates that the length signal helps increase the depth of thinking, enabling the model to engage more in complex reasoning behaviors such as verification, retrospection, and problem decomposition. Such enhanced reflective thinking allows the model to better explore different solution paths and logical turns, thereby improving its ability to handle high-difficulty problems.

To further validate these findings, we conducted a case study by comparing the outputs of the correctness model and format-length model on challenging questions (Appendix Table S1). The format-length model had learned a "step-by-step problem-solving and verification" pattern, which confirmed the effectiveness of our format-length reward mechanism in balancing response length, reasoning depth, and content quality.

Similar to Wang et al. (2025b), we observed that increasing the frequency of reflective language does not necessarily correlate with better model performance. Specifically, models can exhibit over-reflection, characterized by repeatedly switching reasoning paths on complex problems, often leading to failed solutions. This over-reflection is sometimes accompanied by phrase repetition (Appendix Table S2), where models may exploit length rewards through redundancy. We will discuss further in Section A.4.

## 4 DISCUSSION

### 4.1 RETHINKING GROUND TRUTH DEPENDENCY IN MATHEMATICAL REASONING

The remarkable performance of our ground truth-free RL approach begs the question: how can RL without explicit answer supervision match the effectiveness of traditional ground truth-based methods? The answer lies in the latent knowledge already encoded within pre-trained language models. Prior to RL fine-tuning, these models have assimilated vast amounts of knowledge from diverse corpora, enabling them to potentially generate correct answers—RL merely serves as a catalyst to activate this dormant capacity.

Our pass@N experiments provide compelling evidence for this mechanism. By generating N distinct responses per prompt and assessing the presence of correct answers among them, we observe comparable pass@N scores across four conditions: the pre-trained model, the model fine-tuned by GRPO with correctness, format-only, and format-length rewards. As presented in Table 4, which showcases the pass@64 results, we can see that the performance differences between thes methods are relatively minor. The experiments by Yue et al. (2025) also provide similar results. This parity indicates that all RL variants fail to confer new knowledge; instead, they optimize how the model retrieves and structures existing knowledge.

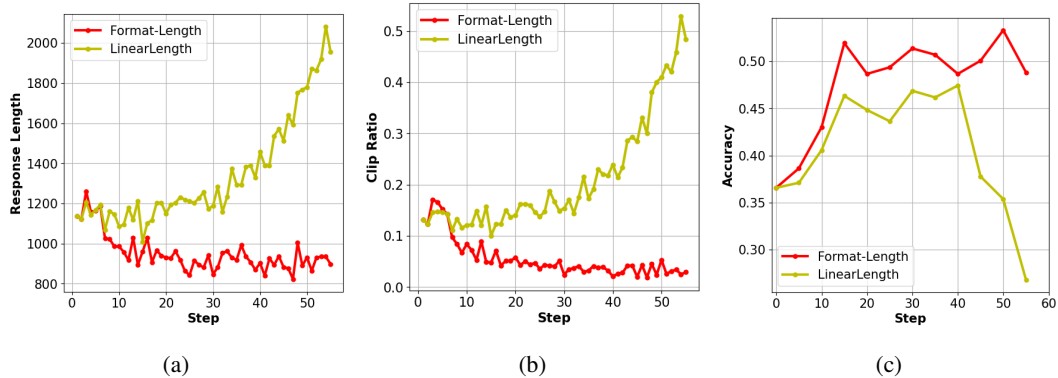

(a)  (b)  (c)

Figure 3: (a) Response length, (b) clip ratio, and (c) average accuracy of benchmark during training.

Table 4: Pass@64 results across different methods.

| Method | AIME2024 | MATH500 | AMC2023 |
|---|---|---|---|
| Qwen-Math-7B | 63.3 | 94.0 | 92.8 |
| Correctness | 73.3 | 94.4 | 90.4 |
| Format-Only | 66.7 | 94.0 | 91.6 |
| Format-Length | 66.7 | 94.4 | 92.8 |

In essence, our findings demonstrate that with the right reward design—such as leveraging format and length cues—RL can effectively stimulate the model's internal reasoning processes. As long as the training mechanism activates the model's latent cognitive abilities, explicit ground truth answers become an optional component rather than an essential requirement for high-performance RL in mathematical reasoning tasks.

## 4.2 FORMAT LEARNING IN RL AND SFT

Since both traditional RL with ground truth rewards and our format-based RL mainly learn answer formatting in the first 15 training steps, a key question arises: how does format learning through RL compare with supervised fine-tuning (SFT)? To answer this question, we carried out a series of comparative experiments, comparing three different training methods: 1) GRPO training with format-based rewards, 2) offline SFT using ground truth chain-of-thought (CoT) examples, and 3) online SFT. Online SFT serves as a middle ground between offline SFT and RL, connecting static supervised learning and the dynamic, feedback-driven RL, which helps us figure out how different training methods affect format learning.

We used Qwen2.5-Math-7B as the original model, which we didn't train, to provide a baseline for comparison. The GRPO(Correctness) was used as a reference to measure the performance of other methods. All experiments were conducted under the setting of sampling from the MATH dataset with a temperature of 0.6.

In the GRPO training with format-based rewards and online SFT experiments, we adopted an online sampling strategy. During training, we constantly sampled model outputs and applied GRPO or SFT based on whether the format was correct. Specifically, online SFT only used format-correct samples to update parameters. All experiments used a batch size of 128 and ran for 100 training steps.

As shown in Table 5, the results offer important insights. Under the temperature=0.6 setting, the GRPO training with format-based rewards and online SFT performed very similarly, achieving comparable format accuracy rates and scores on the MATH500 benchmark. On the other hand, the offline SFT method didn't perform as well, showing lower format accuracy and lower MATH500 scores. These results emphasize the important role of online sampling in making RL more effective for format learning. RL and online SFT can adjust to the quality of real-time outputs, which allows them to optimize answer formatting more efficiently than the static offline SFT. Clearly, the iterative and feedback-driven nature of online training is crucial for quickly improving language models' ability to learn formats.

Table 5: Comparison of format accuracy and answer accuracy across different training methods on the MATH500 benchmark.

| Method | Answer Acc | Format Acc |
|---|---|---|
| Qwen2.5-Math-7B | 61.7 | 87.3 |
| GRPO(Correctness) | 74.0 | 95.0 |
| GRPO(Format-Only) | 70.1 | 96.3 |
| offline SFT | 51.3 | 88.7 |
| online SFT | 71.3 | 95.0 |

### 4.3 DESIGN PRINCIPLES OF FORMAT-LENGTH RL

In the context of language model training, truncation refers to the situation where the generated output exceeds the maximum allowable length (e.g., the context window size of the model) and has to be cut off. Truncation is highly undesirable for several reasons. Firstly, it leads to incomplete responses, which can result in the loss of crucial information and logical steps necessary for correct mathematical reasoning. In the case of mathematical problem-solving, a truncated answer may omit key derivations or final conclusions, rendering the solution incorrect or meaningless. Secondly, truncation can disrupt the coherence and flow of the reasoning process, making it difficult for the model to build on its own arguments and reach a valid conclusion. Prior studies have explored length-based rewards, but their applicability to label-free settings is limited. For example, Yeo et al.Yeo et al. (2025) proposed a cosine-shaped length reward coupled with correctness, while Chen et al. Chen et al. (2025) introduced a linear length reward: $R = L/L_{\text{Max}} + R_{\text{correctness}}$. We reproduced this linear reward and the result is in Figure 3. However, it led to a rapid surge in response length, exceeding the model's context window and causing a 52.9% truncation rate by step 54. This high truncation rate severely degraded performance, as the truncated outputs were often incomplete and lacked the necessary logical structure for accurate mathematical reasoning. This outcome underscores the importance of carefully designing length rewards to balance exploration and efficiency, ensuring that the model generates responses of optimal length without incurring excessive truncation. In contrast, our Format-Length approach maintains a low truncation rate while achieving superior accuracy. By incorporating a length reward that penalizes excessive length before reaching the context limit, our method effectively guides the model to generate concise yet comprehensive responses. This not only prevents reward hacking, where the model might generate overly long or repetitive content to maximize rewards, but also promotes high-quality reasoning, as the model is encouraged to find the most efficient way to express correct mathematical solutions within the given length constraints.

## 5 CONCLUSION

In this study, we analyzed the dynamics of reinforcement learning for large language models in mathematical problem solving. Our experiments show that the early stage of training is largely driven by format learning, where structural feedback alone contributes the majority of performance gains. By complementing this with length-based rewards, we further demonstrated that simple surrogate signals can guide models toward more concise and effective solutions. These findings offer an inspirational perspective on the role of RL: rather than primarily imparting new knowledge, RL may act as a mechanism to activate latent capabilities already embedded in pre-trained models. This suggests that future work should focus on developing lightweight, label-efficient strategies that complement pre-training and enhance the reasoning ability of LLMs in diverse domains.

## 6 LIMITATIONS

There are aspects of our study that merit further exploration. The evaluation of format and length as surrogate signals was predominantly focused on mathematical problem-solving, leaving open the question of their effectiveness in other complex reasoning domains, such as scientific hypothesis testing or advanced programming challenges. Additionally, our experiments were conducted with specific LLM architectures and training configurations, and the performance of this approach may differ when applied to models with varying pre-training paradigms and scale. Noted that our method only works if the chosen base model already has strong latent potential on the target task; If the base model is not powerful, we expect this approach to yield limited gains.

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

# A APPENDIX

## A.1 LLMS USAGE STATEMENT

We just used LLMs (e.g., gpt-4, gpt-5) to review and correct grammar, capitalization, and sentence structure, primarily for Section 1 and the appendix. We also carefully reviewed the LLM's suggested revisions, rejected any inappropriate suggestions, and adjusted them to improve the content's readability.

## A.2 ETHICS STATEMENT

This work does not involve human subjects, sensitive or private data, or high-risk applications. All datasets are either publicly available or synthetically generated, and no ethical concerns arise regarding bias, fairness, privacy, or safety. We declare no conflicts of interest, and conclude that our paper raises no ethical issues.

## A.3 RELATED WORK

RL has been proven effective in enhancing LLM performance. PPO Schulman et al. (2017) and GRPO Shao et al. (2024) are widely used in RL frameworks for LLMs, with detailed introductions provided in Appendix A.5. Recent research uses scaled-up RL training to enable LLMs to explore reasoning paths for complex problems. For example, DeepSeek-R1 DeepSeek-AI et al. (2025) achieved excellent results in math and coding tasks through large-scale RL on an unsupervised base model, without relying on pre-trained reward models or MCTS. Kimi-k1.5 Team et al. (2025) enhances general reasoning via RL, focusing on multimodal reasoning and controlling thinking length.

**Format reward in RL.** DeepSeek-R1 DeepSeek-AI et al. (2025) uses format rewards to structure model outputs. Liu et al. Liu et al. (2025a) noted format rewards dominate early training. Our study isolates the influence of answer rewards and designs a format for math reasoning tasks. Experiments show using our format in early RL training matches performance of answer reward training.

**Length Control in RL.** DeepSeek-R1 DeepSeek-AI et al. (2025) found response length and evaluation metrics increase with RL training steps until an "Aha moment". Other studies explore length reward functions' impacts. Yeo et al. Yeo et al. (2025) observed response lengths decline due to model size and KL divergence penalties. Chen et al. Chen et al. (2025) argued direct length extension training may harm performance. In contrast, our length reward penalizes overly long responses, guiding concise outputs. Experiments show combining length and format rewards outperforms answer rewards.

**Label-Free RL.** Recent advances in *Label-Free Reinforcement Learning with Verifiable Rewards (RLVR)* have sought to eliminate the dependency on large-scale human-labeled datasets by leveraging alternative signals. Early work such as Jiao et al. (2025) and Prasad et al. (2025) demonstrated that pseudo-feedback and consistency-based objectives can serve as substitutes for explicit preference labels. Building on this idea, Zhao et al. (2025a) and Zhang et al. (2025b) explored fully unsupervised or self-play paradigms, showing that reasoning can be improved purely through self-generated questions, answers, and verification signals. Alternative signals such as entropy minimization and confidence regularization were further investigated in works like Li et al. (2025a) , while Zhang et al. (2025a) and Zhao et al. (2025b) leveraged path stability and internal self-correction as reward surrogates. In mathematical domains, Xiong et al. (2025) showed that structural constraints and length-based signals can significantly boost performance without ground truth labels. Cross-modal studies such as Ding & Zhang (2025) extend these ideas to audio-visual reasoning. Collectively, these works establish a spectrum of label-free reward designs—ranging from pseudo-preference, entropy-based confidence, format/length surrogates, to self-play—that consistently improve reasoning performance across mathematics, coding, and multimodal tasks while raising new challenges in evaluation reliability and reward hacking prevention.

## A.4 MITIGATING REPETITION AND REWARD HACKING

A potential concern with length-based rewards is the risk of reward hacking, where the model generates repetitive content to increase its length. To address this, we employed the longest repeated

substring analysis to measure repetition. The longest repeated substring ratio (Figure S1) provides a normalized perspective on repetition. At the start of training, both the format-length and correctness models exhibited high levels of repetition, mainly due to incorrect formatting issues, such as stacked instances of '\\boxed'. However, this problem was resolved after just 15 training steps. The repetition rate then dropped significantly and remained stable throughout the subsequent training process. These findings demonstrate that the format-length reward mechanism effectively balances response length, reasoning depth, and content quality. By integrating format and length signals, our approach not only improves performance on mathematical reasoning tasks but also mitigates the risks associated with traditional length-based rewards, like repetition and reward hacking.

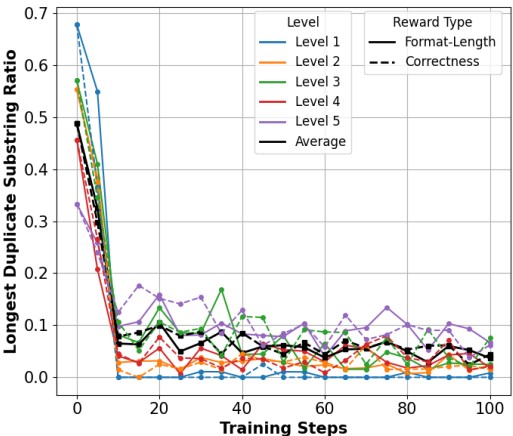

Figure S1: Longest duplicate substring ratio of MATH500 evaluation benchmark during training.

## A.5 INTRODUCION OF PPO AND GRPO

### A.5.1 PROXIMAL POLICY OPTIMIZATION

PPO is a widely-used and highly effective algorithm in the field of RL. At its core, PPO aims to optimize the policy of an agent to maximize the expected cumulative reward over time. The algorithm is based on the policy gradient method, which updates the policy by computing the gradient of the expected reward with respect to the policy parameters. The key idea behind PPO is to balance the trade-off between exploration and exploitation during the policy update process. It does this by introducing a clipped surrogate objective function. Let $\pi_\theta$ be the policy parameterized by $\theta$, and $\pi_{\theta_{old}}$ be the old policy. Given a set of trajectories collected from the environment, the objective of PPO is to maximize the following clipped objective function:

$$\mathbb{E}_{\pi_{\theta_{old}}} \left[ \min \left( r_t(\theta) A_t, \text{clip}(r_t(\theta), 1 - \epsilon, 1 + \epsilon) A_t \right) \right] \tag{6}$$

where

$$r_t(\theta) = \frac{\pi_\theta(a_t|s_t)}{\pi_{\theta_{old}}(a_t|s_t)} \tag{7}$$

is the probability ratio of the new policy $\pi_\theta$ to the old policy $\pi_{\theta_{old}}$ for taking action $a_t$ in state $s_t$, $A_t$ is the advantage function that estimates how much better an action is compared to the average action in state $s_t$, and $\epsilon$ is a hyperparameter that controls the clipping range. The clipping operation ensures that the policy update is not too drastic, preventing the policy from diverging significantly from the old policy in a single update step.

To compute the advantage function $A_t$, PPO typically relies on value function estimation combined with Generalized Advantage Estimation (GAE). The value function $V(s)$ parameterized by $\phi$, predicts the expected cumulative reward from state $s$. It is trained via temporal difference (TD) learning to minimize the squared error:

$$L^{\text{Value}}(\phi) = \mathbb{E} \left[ \left( V_\phi(s_t) - (R_t + \gamma V_\phi(s_{t+1})) \right)^2 \right], \tag{8}$$

where $R_t$ is the reward given by a reward model or a reward function and $\gamma$ is the discount factor. The advantage $A_t$ is then calculated using GAE, which generalizes multi-step TD errors with a tunable parameter $\lambda \in [0,1]$ to balance bias and variance:

$$A_t^{\text{GAE}(\gamma,\lambda)} = \sum_{l=0}^{\infty} (\gamma\lambda)^l \delta_{t+l},$$

$$\delta_t = R_t + \gamma V(s_{t+1}) - V(s_t). \tag{9}$$

Here, $\lambda = 0$ reduces to single-step TD error, while $\lambda = 1$ recovers Monte Carlo advantage estimation. By integrating GAE, PPO efficiently utilizes trajectory data while maintaining stable policy updates.

### A.5.2 GROUP RELATIVE POLICY OPTIMIZATION

GRPO is an efficient reinforcement learning algorithm that improves upon PPO by eliminating the need for a separate value function. GRPO estimates advantages through group-relative normalization: for a given input query $q$, the behavior policy $\pi_{\theta_{old}}$ samples $G$ responses $\{o_i\}_{i=1}^G$, then calculates each response's advantage as:

$$A_t^{\text{GRPO}}(o_i) = \frac{R(o_i) - \text{mean}(\{R(o_j)\}_{j=1}^G)}{\text{std}(\{R(o_j)\}_{j=1}^G)}, \tag{10}$$

where $R(o_i)$ is the reward of response $o_i$.

### A.6 EVALUATION DETAILS

We used `vllm` for inference with greedy decoding (temperature = 0) to ensure reproducibility. Since vLLM's batched inference produces different outputs for the same input under different batch sizes, we set the validation batch size to 128 and evaluate each dataset independently to ensure consistency in evaluation. Because we used the Qwen2.5-Math base models with a context length of 4k, we set the generation budget for all compared baselines to 3k.

### A.7 TEMPLATE

---

**Qwen-Math Template**

```
<|im_start|>system
```
Please reason step by step, and put your final answer within `\boxed{}`. `<|im_end|>`
```
<|im_start|>user
{question}
<|im_end|>
<|im_start|>assistant
```

---

**Deepseek-R1 Template**

A conversation between User and Assistant. The User asks a question, and the Assistant solves it. The Assistant first thinks about the reasoning process in the mind and then provides the User with the answer. The reasoning process is enclosed within `<think> </think>` and the answer is enclosed within `<answer> </answer>` tags, respectively, i.e., `<think>` reasoning process here `</think>` `<answer>` answer here `</answer>`.
User: {question}
Assistant:

---

### A.8 DETAILED FORM OF FORMAT REWARD

DeepSeek-R1 DeepSeek-AI et al. (2025) introduced a format reward to assess whether the model's output aligns with the Deepseek-R1 template(Appendix A.7) format (i.e., writing the reasoning process within `<think> </think>` tags and placing the answer within `<answer></answer>`

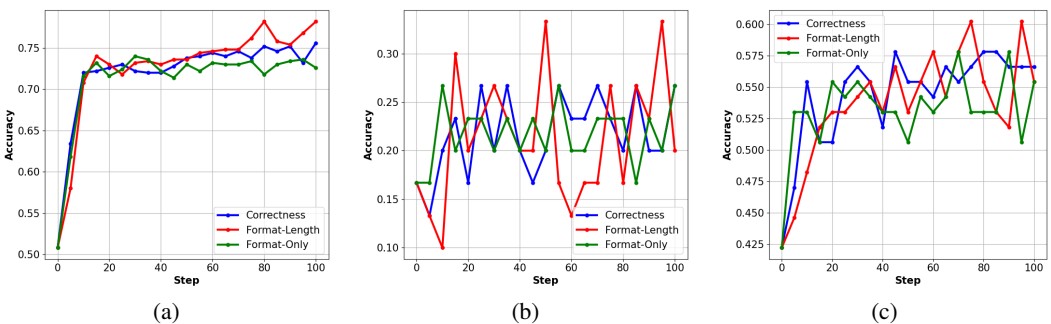

Figure S2: Accuracy curves on (a) MATH500, (b) AIME2024, and (c) AMC2023 benchmarks training on the **DeepScaleR**.

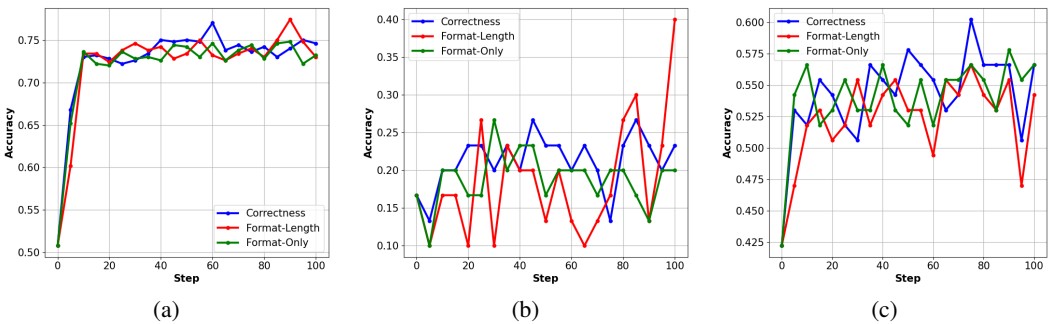

Figure S3: Accuracy curves on (a) MATH500, (b) AIME2024, and (c) AMC2023 benchmarks training on **Math-train** dataset.

tags). Our format reward is designed based on Qwen-Math template(in Appendix A.7), which evaluates whether the model's output contains \boxed{} and checks whether the content inside the \boxed{} is valid mathematical content. A specific example is shown below:

```
Right case:
    \boxed{1}
    \boxed{\frac{3}{2}}
    \boxed{x^2 + 12y =1}
    \boxed{(0,\infty)}
Wrong case:
    \boxed{}
    \boxed{x +* 2}
    \boxed{(0,1 }
```

### A.9 PERFORMANCE CURVES OF DIFFERENT REWARD TRAININGS ON BENCHMARKS

See Figure S2 and S3.

### A.10 CASE STUDY

See Table S1 and S2.

**PROMPT:** system\nPlease reason step by step, and put your final answer within \\boxed{}. \n user \n You have seven bags of gold coins. Each bag has the same number of gold coins. One day, you find a bag of 53 coins. You decide to redistribute the number of coins you have so that all eight bags you hold have the same number of coins. You successfully manage to redistribute all the coins, and you also note that you have more than 200 coins. What is the smallest number of coins you could have had before finding the bag of 53 coins?\nassistant\n

**RESPONSE_FORMAT-LENGTH:** To solve this problem, we need to determine the smallest number of coins in each of the original seven bags such that when we add the 53 coins from the new bag, the total number of coins can be evenly distributed among eight bags, and the total number of coins is more than 200.\n\nLet's break down the problem step by step :\n\n1. **Define Variables:**\n - Let x be the number of coins in each of the original seven bags.\n - The total number of coins in the original seven bags is 7x .\n - After finding the new bag with 53 coins, the total number of coins becomes 7x + 53 .\n\n2. **Redistribution Condition:**\n - We need to redistribute these coins so that each of the eight bags has the same number of coins.\n - Therefore , 7x + 53 must be divisible by 8.\n\n3. **Mathematical Formulation:**\n - We need to find the smallest x such that 7x + 53 \\equiv 0 \\ pmod{8} .\n - This can be rewritten as 7x \\equiv -53 \\pmod{8} .\n - Since -53 \\equiv 3 \\pmod{8} (because -53 + 56 = 3 and 56 is a multiple of 8), we have 7x \\equiv 3 \\pmod{8} .\n\n •••••• 6. **Calculate the Total Number of Coins:**\n - If x = 5 , the total number of coins in the original seven bags is 7 \\times 5 = 35 .\n - Adding the 53 coins from the new bag, the total number of coins is 35 + 53 = 88 .\n\n7. ** Check the Condition:**\n - We need to ensure that the total number of coins is more than 200.\n - Since 88 is not more than 200, we need to find the next smallest x that satisfies x \\equiv 5 \\pmod{8} .\n\n8. **Next Smallest x :**\n - The next smallest x is x = 5 + 8 = 13 .\n - If x = 13 , the total number of coins in the original seven bags is 7 \\times 13 = 91 .\n - Adding the 53 coins from the new bag, the total number of coins is 91 + 53 = 144 .\n\n9. ** Continue Checking :**\n - Since 144 is still not more than 200, we continue to the next smallest x .\n - The next smallest x is x = 13 + 8 = 21 .\n - If x = 21 , the total number of coins in the original seven bags is 7 \\times 21 = 147 .\n - Adding the 53 coins from the new bag, the total number of coins is 147 + 53 = 200 .\n\n10. ** Final Check :**\n - Since 200 is not more than 200, we need to find the next smallest x .\n - The next smallest x is x = 21 + 8 = 29 .\n - If x = 29 , the total number of coins in the original seven bags is 7 \\times 29 = 203 .\n - Adding the 53 coins from the new bag, the total number of coins is 203 + 53 = 256 .\n\n11. **Conclusion:**\n - The smallest number of coins in each of the original seven bags that satisfies all conditions is x = 29 .\n - Therefore , the smallest number of coins you could have had before finding the bag of 53 coins is 7 \\times 29 = 203 .\n\nThus, the final answer is:\n\\[\n\\**boxed{203}** \n\\]. ✔

**RESPONSE_CORRECTNESS:** To solve this problem, let's denote the number of coins in each of the original seven bags as x . Therefore , the total number of coins you originally had is 7x . After finding the bag of 53 coins, you now have 7x + 53 coins. You redistribute these coins into eight bags, so each bag now has frac7x + 53{8} coins. Since the number of coins in each bag must be an integer, 7x + 53 must be divisible by 8. Additionally, we know that the total number of coins after redistribution is more than 200, so 7x + 53 > 200 . This simplifies to 7x > 147 , or x > 21 . We need to find the smallest integer x such that 7x + 53 is divisible by 8 and x > 21 . Let's iterate through the possible values of x starting from 22 and check the conditions.\n```python\nIterate through possible values of x starting from 22\nfor x in range(22, 1000):\n total_coins = 7 * x + 53\n if total_coins % 8 == 0:\n smallest_number_of_coins = 7 * x\n break\n\nprint(smallest_number_of_coins)\n ```\n```output\n183\n```\nThe smallest number of coins you could have had before finding the bag of 53 coins is **boxed{183}**. ✖

Table S1: Model output on complex mathematical problems after training on format-length reward and correctness reward. The blue boxes in the figure represent step-by-step problem-solving behavior, while the red boxes indicate verification behavior. Clearly, both types of behavior occur far more frequently in the format-length model than in the correctness model. As for the results, the former produces correct answers, whereas the latter produces incorrect ones.

**PROMPT:** system\nPlease reason step by step, and put your final answer within \\boxed{}.\n user \nDenali and Nate work for a dog walking business and are paid for each dog they walk. Denali is responsible for 16 dogs and Nate is responsible for 12 dogs. Under the company's new policy, they will be assigned or unassigned new dogs in groups of x dogs. The ratio of Denali's pay to Nate's pay would be the same if Denali started walking 4x more dogs and Nate stayed at 12 dogs or if x of Nate's dogs were reassigned to Denali. Find x if x\\neq0.\nassistant\n,

**RESPONSE:** Let's denote the payment per dog as \\( p \\). The initial ratio of Denali's pay to Nate's pay is \\(\\frac16p12p = \\frac43\\).\n\nIf Denali starts walking \\(4x\\) more dogs and Nate stays at 12 dogs, the new ratio of their pays would be \\(\\frac{(16 + 4x)p}{12p} = \\frac{16 + 4x}{12}\\). $\boxed{\text{Since}}$ this ratio is equal to \\(\\frac43\\), we can set up the equation:\n\\[\n\\frac{16 + 4x}{12} = \\frac43.\n\\]\nCross-multiplying gives:\n\\[3(16 + 4x) = 4 \\cdot 12,\n\\]\n\\[48 + 12x = 48,\n\\]\n\\[12x = 0,\n\\]\n\\[x = 0.\n\\]\n $\boxed{\text{Since}}$ \\(x \\neq 0\\), we need to consider the second scenario where \\(x\\) of Nate's dogs are reassigned to Denali. In this case, the new number of dogs Denali is responsible for is \\(16 + x\\) and the new number of dogs Nate is responsible for is \\(12 - x\\). The new ratio of their pays would be \\(\\frac{(16 + x)p}{(12 - x)p} = \\frac{16 + x}{12 - x}\\). $\boxed{\text{Since}}$ this ratio is equal to \\(\\frac43\\), we can set up the equation:\n\\[\n\\frac{16 + x}{12 - x} = \\frac43.\n\\]\nCross-multiplying gives:\n\\[3(16 + x) = 4(12 - x),\n\\]\n\\[48 + 3x = 48 - 4x,\n\\]\n\\[7x = 0,\n\\]\n\\[x = 0.\n\\]\nAgain, $\boxed{\text{since}}$ \\(x \\neq 0\\), we need to $\boxed{\text{recheck}}$ the problem constraints. The only way the ratio can remain the same is if the number of dogs reassigned is such that the ratio of the new number of dogs is the same as the original ratio. Let's solve it again by checking the constraints properly.\n\nThe correct approach is to solve the equation \\(\\frac{16 + x}{12 - x} = \\frac43\\) properly.

> Let's solve it $\boxed{\text{step by step}}$ again.\n\n\\[3(16 + x) = 4(12 - x), \n\\]\n\\[48 + 3x = 48 - 4x,\n\\]\n\\[7x = 0,\n\\]\n\\[x = 0.\n\\]\n $\boxed{\text{Since}}$ \\(x \\neq 0\\), we need to consider the possibility that the problem might have a different interpretation or constraint. Let's $\boxed{\text{recheck}}$ the problem statement and constraints. The only way the ratio can remain the same is if the number of dogs reassigned is such that the ratio of the new number of dogs is the same as the original ratio. Let's solve it again by checking the constraints properly.

> **Repeat1:** Let's solve it $\boxed{\text{step by step}}$ again.\n\n\\[3(16 + x) = 4(12 - x), \n\\]\n\\[48 + 3x = 48 - 4x,\n\\]\n\\[7x = 0,\n\\]\n\\[x = 0.\n\\]\n $\boxed{\text{Since}}$ \\(x \\neq 0\\), we need to consider the possibility that the problem might have a different interpretation or constraint. Let's $\boxed{\text{recheck}}$ the problem statement and constraints. The only way the ratio can remain the same is if the number of dogs reassigned is such that the ratio of the new number of dogs is the same as the original ratio. Let's solve it again by checking the constraints properly.

> **Repeat2:** Let's solve it $\boxed{\text{step by step}}$ again.\n\n\\[3(16 + x) = 4(12 - x), \n\\]\n\\[48 + 3x = 48 - 4x,\n\\]\n\\[7x = 0,\n\\]\n\\[x = 0.\n\\]\n $\boxed{\text{Since}}$ \\(x \\neq 0\\), we need to consider the possibility that the problem might have a different interpretation or constraint. Let's $\boxed{\text{recheck}}$ the problem statement and constraints. The only way the ratio can remain the same is if the number of dogs reassigned is such that the ratio of the new number of dogs is the same as the original ratio. Let's solve it again by checking the constraints properly.

......

Table S2: Over-reflection in sentence repetition. The red boxes in the figure indicate reflective words, while the blue boxes represent repeated phrases (with the numbers indicating the frequency of repetition). It can be observed that reflective words appear within the repeated phrases.

