# OpenReview forum: "Surrogate Signals from Format and Length: Reinforcement Learning for Solving Mathematical Problems without Ground Truth Answers"
_ICLR.cc/2026/Conference — Submitted to ICLR 2026_

### Official Review · Reviewer_6Aps · 2025-10-30

**Soundness:** 3
**Presentation:** 2
**Contribution:** 2
**Rating:** 4
**Confidence:** 4

**Summary:**

The paper studies math reasoning and uses two proxy signals (i.e., format and length), to replace answer-based rewards for label-free RL. These signals are added via replacing GRPO algorithm's GT reward to the combination of these two proxy rewards.
1. The format reward R_f is a yes/no check: did the model follow the required template (e.g., did it include a valid boxed final answer).
2. The length reward R_l is a piecewise “rise-then-fall” function that encourages a moderate chain of though (not too long nor too short).

Experiments show that early gains mostly come from learning the right format; within about 15 steps the model gets most of the accuracy improvement. After that, adding the length signal gives extra gains. Across several base models and datasets, this approach matches or slightly beats answer-based GRPO. The authors argue that RL here mainly activates reasoning ability already present from pretraining, rather than emerging new knowledge.

**Strengths:**

1. This paper provides a concrete, clear, reproducible reward design.
2. The paper empirically shows a two-phase pattern: (1) early gains from learning the output format (2) followed by further gains from length control
3. The paper is in general lean presented, very easy-to-follow, (e.g., Qwen-Math prompt format, format checker, GRPO setup) and the formulas are also well-organized.

**Weaknesses:**

1. Current math-QA benchmarks already provide relatively rich verifiable signals; harder settings (e.g., proof generation, open-ended reasoning) are underexplored so far. So I do hope the authors can extend the experiments to these tasks.

2. Results look Qwen-centric with unclear generalization to Qwen3/Llama/Mistral. In fact, there are several papers showing that qwen2.5 series of models can be successfully "RL"ed even with spurious rewards. I really want to see results on more models.

3. Will these reward function design also have potential for over-reflection, repetition, and mechanical length inflation? i.e., How to de-risk these factors.

**Questions:**

See weakness.

---

> ### Author Response · Authors · 2025-11-25
> **The 1/2 of the reply**
>
> Thank you for accurately identifying the issues in our paper. Please allow me to respond in detail.
>
> ---
> ## W1 Generalization
>
> Thank you for your valuable insights. We fully agree that extending surrogate signals to tasks such as **proof generation** and **open-ended reasoning** is a highly meaningful direction.
> However, it is important to note that different domains require **different forms of surrogate signals**—for example, open-ended reasoning requires rubric-based and rule-based verification. We plan to explore these areas further in future work.
>
> That said, we additionally evaluated our method on several mainstream code-generation benchmarks ( HumanEvalPlus, MBPPPlus, CruxEval ) and GPQADiamond—using **Qwen2.5-Math-7B**. Since our training data contains only mathematics, all these benchmarks are **OOD (out-of-distribution)**.
> From the results below, our method surpasses the baseline across OOD datasets Avg , especially on **CruxEval (Output)**, the most challenging of them. This provides strong evidence of the **generalization ability** of our approach.
>
> | Method                  | GPQADiamond | HumanEvalPlus | MBPPPlus | CruxEval(Output) | Avg. |
> |-------------------------|-------------|---------------|----------|------------------|------|
> | Qwen-Math-7B         | 24.2        | 52.4          | 40.5     | 27.5             | 36.2 |
> | Correctness             | 25.8        | 51.8          | **52.6** | 20.9             | 37.8 |
> | Format-Only             | 25.3        | **62.2**      | 50.5     | 32.8             | 42.7 |
> | Format-Length (p=0.8)   | **29.8**    | 52.4          | 50.0     | **57.6**         | **47.5** |
>
> ---
>
> ## W2 Universality
>
> To demonstrate the **universality** of our method, we conducted experiments on **three base models with different architectures**:
> - Qwen3-30B-A3B-Base
> - Mathstral-7B-v0.1
> - deepseek-math-7b-base
>
> On **deepseek-math-7b-base**, our Format+Length method **outperformed** the baseline.
> On **Mathstral-7B-v0.1 (Mistral)**, our Format+Length method reached **96%** of the baseline performance.
>
> To reduce data contamination, experiments on **Qwen3-30B-A3B-Base** additionally used **three newly decontaminated datasets** (see §3.3.1 of the paper). With these, our Format+Length method again reached **96%** of the baseline.
>
> Regarding the use of “fake rewards,” the Qwen 2.5 series can still successfully undergo “reinforcement learning.” This supports our central argument:
>
> 1. **Pretraining already teaches models mathematical knowledge.**
>    RL mainly *activates* the knowledge that the model already possesses, rather than overwriting it.
>
> 2. **Fake rewards are not strong enough to override pretrained distributions.**
>    They cannot force the model to “forget” its knowledge, but can still guide optimization.
>
> Moreover, prior work[1] suggests that Qwen models may contain **data contamination**, which could partially explain the effectiveness of fake rewards.
> Thus, for Qwen-based experiments, we incorporated decontaminated datasets and our method still achieved strong results.
>
> ### deepseek-math-7b-base
>
> | Method                 | AIME2024 | MATH500 | AMC2023 | Avg. |
> |-----------------------|----------|---------|---------|------|
> | Deepseek-Math-7B-Base | 0.0      | 1.0     | 0.0     | 0.3  |
> | Correctness           | 0.0      | 22.4    | **16.9**| 13.1 |
> | Format-Only           | 0.0      | **27.4**| 12.1    | 13.2 |
> | Format-Length         | **3.3**  | 25.8    | 12.1    | **13.7** |
>
> ### Mistral-7B(Mathstral-7B-v0.1)
>
> | Method                       | AIME2024 | MATH500 | AMC2023 | Avg. |
> |-----------------------------|----------|---------|---------|------|
> | Mistral-7B(Mathstral-7B-v0.1)  | 0.0      | 51.4    | 19.3    | 23.6 |
> | Correctness                 | **6.7**  | 61.8    | 30.1    | **32.9** |
> | Format-Only                 | 3.3      | 52.0    | 27.7    | 27.7 |
> | Format-Length               | 0.0      | **62.8**| **32.5**| 31.8 |
>
> ### Qwen3-30B-A3B-Base
>
> | Method                  | AIME2025 | MINERVA | LIVE_MATH_BENCH2025 | AIME2024 | AMC2023 | MATH500 | Avg. |
> |------------------------|----------|---------|-------------|----------|---------|---------|------|
> | Qwen3-30B-A3B-Base     | 10.0     | 14.3    | 9.1         | 20.0     | 41.0    | 70.0    | 27.4 |
> | Correctness            | **13.3** | **24.6**| 13.1        | **26.7** | **60.2**| 80.4    | 36.4 |
> | Format-Only            | **13.3** | 23.9    | **14.1**    | 20.0     | 56.6    | 80.0    | 34.7 |
> | Format-Length          | **13.3** | 23.9    | 12.1        | **26.7** | 53.0    | **81.2**| 35.0 |
>
> (*All Qwen3-30B-A3B-Base runs experienced noticeable training instability. We suspect that the MoE routing mechanism interacts poorly with GRPO’s clipping effects.*)
>
> ---
> [1] Mingqi Wu, Zhihao Zhang, et al. "Reasoning or Memorization? Unreliable Results of Reinforcement Learning Due to Data Contamination"

---

> ### Author Response · Authors · 2025-11-25
> **The 2/2 of the reply**
>
> ## W3
>
> Your concern is very reasonable, and we considered it as well.
> In Appendix **A.4**, we report the **Longest duplicate substring ratio** during training.
> Compared with the baseline (Correctness reward), the **Format+Length** method does **not** increase repetition, demonstrating that our method: improves mathematical reasoning performance,  while **without** introducing the well-known risks associated with length-based rewards.

---

### Official Review · Reviewer_XzHt · 2025-11-01

**Soundness:** 3
**Presentation:** 3
**Contribution:** 2
**Rating:** 4
**Confidence:** 4

**Summary:**

This paper proposes using simpler losses that only account for the length and formatting of the answers, as opposed to (just) the correctness, in order to boost RL performance.

**Strengths:**

The main strengths of this paper:
- the losses proposed by the authors do not require ground truth labels and are simple and interpretable
- the proposed method achieves a large boost quite quickly in training, showing that a lot of performance can be extracted by supervising the formatting and length

**Weaknesses:**

The main weaknesses of this paper:
- In general, I am not sure supervising just for formatting and length would be enough on more complicated math proofs/code, and more importantly, much longer traces
- I am not sure exactly what the main message of this paper is meant to be. The authors' experiments show that rewarding for correctness achieves at least the same performance as rewarding for formatting and length on these tasks (for example, format-only and format-length in Table 2 are much worse than correctness for the Llama model). I am not sure whether the authors are suggesting that correctness rewards should be replaced with rewards for formatting/length, or that both should be used concurrently
- The authors claim that format length can surpass correctness check (in the abstract), and while this is true in Table 2, gaps in post training of roughly 2-3% are not clearly statistically significant given the very noisy nature of RL
- How does one tune the parameter p controlling the length of the chain of thought?

**Questions:**

What is meant by online SFT? Does it mean that the model is SFTd on its own generations, similar to Expert Iteration (EI)?
Have you swept the hyperparameters for GRPO and SFT?
Are the models that get low reward actually not solving the problems, or is it the case that simply they are not wrapping their answer in \boxed?

---

> ### Author Response · Authors · 2025-11-25
> **The 1/2 of the reply**
>
> Thank you for your insightful and constructive review. Below, we first address the specific questions raised in the *Questions* section.
>
> Yes, **online SFT** in our paper refers to supervised fine-tuning on the model’s own self-generated responses, and these responses are selected using a format-based reward. As reviewer *CwVY* correctly pointed out, this serves as an ablation study: online SFT achieves results very similar to GRPO with format rewards, demonstrating that **the effectiveness of format learning is not tied to the GRPO algorithm itself**.
>
> In this experiment, we did not perform hyperparameter searches for Online/Offline SFT or GRPO because the performance trend is already very clear. Additionally, regarding the poor performance of offline SFT, Table 5 in Section 4.2 shows that **its format correctness (e.g., producing a `\boxed{}` answer) is similar to online SFT**, while the correctness gap is large. This indicates that the performance drop is *not* primarily caused by incorrect answer formatting.
>
> ---
>
>
> ## **W1. Generalization and Response Length**
>
> We additionally evaluated several mainstream code-generation benchmarks (HumanEvalPlus, MBPPPlus, CruxEval) and GPQADiamond, using **Qwen2.5-Math-7B**. Since our training data consist purely of mathematical content, these are **out-of-distribution (OOD)** tasks.
>
> The table below shows that our method outperforms the baseline on OOD datasets, especially on the most challenging benchmark, **CruxEval (Output)**. This strongly supports the **generalization capability** of our approach.
>
> | Method                | GPQADiamond | HumanEvalPlus | MBPPPlus | CruxEval(Output) | Avg.     |
> | --------------------- | ----------- | ------------- | -------- | ---------------- | -------- |
> | Qwen2.5-Math-7B       | 24.2        | 52.4          | 40.5     | 27.5             | 36.2     |
> | Correctness           | 25.8        | 51.8          | **52.6** | 20.9             | 37.8     |
> | Format-Only           | 25.3        | **62.2**      | 50.5     | 32.8             | 42.7     |
> | Format-Length (p=0.8) | **29.8**    | 52.4          | 50.0     | **57.6**         | **47.5** |
>
> Our main experiments use **Qwen2.5-Math-7B**, whose pretraining context length is 4k tokens [1]. Although it is technically possible to extend the context length via RoPE scaling, this may introduce instability. We also deliberately avoid Instruction-tuned models because our focus is on **training from scratch on base models**, similar in spirit to *DeepSeek-R1-Zero*. All RLVR papers we compare against (LIMR, SimpleRL, Oat-Zero) use a comparable context length.
>
> ---
>
> ## **W2. Effectiveness of Surrogate Rewards**
>
> Our claim is that surrogate signals can approximate or even surpass true correctness rewards in certain scenarios.
> To study this, we added an experiment combining Correctness + Format-Length:
>
> ```python
> if correctness_reward > 0:
>     score = correctness_reward + (format_reward + length_reward) * 0.2
> else:
>     if format_reward > 0:
>         score = (format_reward + length_reward) * 0.1
>     else:
>         score = min(0.0, length_reward)
> ```
>
> The intuition is:
>
> * when the model answers correctly, we upweight the correctness reward;
> * when the answer is incorrect but the format is valid, we provide a small length-based reward;
> * when both correctness and format are invalid, we penalize length (especially if overlong).
>
> **Results on Qwen2.5-Math-7B:**
>
> | Method                       | AIME2024 | MATH500 | AMC2023 | Avg. |
> | --------------------------- | -------- | ------- | ------- | ---- |
> | Format-Length               | 33.3     | 33.3    | 60.2    | 56.8 |
> | Correctness + Format-Length | 26.7     | 78.2    | 61.5    | 55.4 |
>
> The addition of correctness reward **did not improve performance**, showing that **Format-Length rewards already capture most of the gains**.

---

> ### Author Response · Authors · 2025-11-25
> **The 2/2 of the reply**
>
> We acknowledge that the improvements on Llama models are weaker than on Qwen. However, as discussed in Section 3.3, our Format+Length signal achieves **92.2%** of correctness reward performance on Llama—remarkable given that it uses *zero labeled data*. This strongly supports the view that **RL does not introduce much new knowledge**, but instead **activates knowledge already present from pretraining**.
>
> To demonstrate cross-architecture generality, we further tested on other mathematically pre-trained base models:
>
> ### **DeepSeek-Math-7B-Base**
>
> | Method                 | AIME2024 | MATH500  | AMC2023  | Avg.     |
> | --------------------- | -------- | -------- | -------- | -------- |
> | Deepseek-Math-7B-Base | 0.0      | 1.0      | 0.0      | 0.3      |
> | Correctness           | 0.0      | 22.4     | **16.9** | 13.1     |
> | Format-Only           | 0.0      | **27.4** | 12.1     | 13.2     |
> | Format-Length         | **3.3**  | 25.8     | 12.1     | **13.7** |
>
> On **deepseek-math-7b-base**, our method surpasses the baseline.
>
> ### **Mistral-7B(Mathstral-7B-v0.1)**
>
> | Method             | AIME2024 | MATH500  | AMC2023  | Avg.     |
> | ----------------- | -------- | -------- | -------- | -------- |
> | Mathstral-7B-v0.1 | 0.0      | 51.4     | 19.3     | 23.6     |
> | Correctness       | **6.7**  | 61.8     | 30.1     | **32.9** |
> | Format-Only       | 3.3      | 52.0     | 27.7     | 27.7     |
> | Format-Length     | 0.0      | **62.8** | **32.5** | 31.8     |
>
> Format+Length reaches **96%** of correctness reward performance on Mathstral, further supporting general applicability.
>
> ---
>
> ## **W3. Surrogate Rewards Achieve >90% of Correctness Performance**
>
> While we agree that a 2–3% gap is effectively noise, across *all* model families, our Format-Length reward achieves **>90%** of correctness-based RL performance.
>
> | Model                        | Init | Correctness   | Format-Length | Format-Length / Correctness (%) |
> | ---------------------------- | ---- | ------------- | ------------- | ------------- |
> | Qwen-Math-1.5B               | 27.1 | 43.1 (↑59%)   | 43.5 (↑60%)   | 100.9%        |
> | Qwen-Math-7B                 | 36.6 | 53.0 (↑45%)   | 56.8 (↑55%)   | 107.2%        |
> | Qwen-Math-72B                | 56.2 | 64.5 (↑15%)   | 62.7 (↑12%)   | 97.2%         |
> | Llama3.1-8B (OctoThinker-8B) | 18.7 | 38.7 (↑107%)  | 35.7 (↑91%)   | 92.2%         |
> | Deepseek-Math-7B-Base        | 0.3  | 13.1 (↑4267%) | 13.7 (↑4467%) | 104.6%        |
> | Mistral-7B(Mathstral-7B-v0.1)    | 23.6 | 32.9 (↑39%)   | 31.8 (↑35%)   | 96.7%         |
>
> (*The scores in the table are the average of the three evaluation sets (AIME2024, MATH500, AMC2023).Values in parentheses indicate improvement relative to the initial model.*)
>
>
>
> For a method requiring **no labeled data**, the Format-Length reward is already strong enough. Moreover, it again demonstrates that RL **activates existing knowledge** rather than learning new mathematical knowledge.
>
> ---
>
> ## **W4. Sensitivity to the Parameter p**
>
>
> In the paper, the default value is *p = 0.5*. We additionally tested *p = 0.4, 0.6, 0.8*:
>
> | Method   | AIME2024 | MATH500 | AMC2023 | Avg.      |
> | ------- | -------- | ------- | ------- | -------- |
> | p = 0.5 | 33.3     | 76.8    | 60.2    | 56.8     |
> | p = 0.4 | 33.3     | 77.2    | 59.0    | 56.5     |
> | p = 0.6 | 30.0     | 75.8    | 60.2    | 55.4     |
> | p = 0.8 | 33.3     | 78.6    | 60.2    | **57.4** |
>
> All three variants perform similarly to *p = 0.5*, with *p = 0.8* being slightly better. This suggests the method is **not sensitive** to the choice of *p*.
>
> We also tested extreme values:
>
> * **p = 1.0** (no penalty for long responses):
>   similar to the *LinearLength* setting discussed in Sec. 4.3—response length quickly approaches the context limit, causing truncation.
> * **p = 0.1**:
>   response lengths shrink throughout training, limiting exploration and eventually causing a decline after an initial spike.
>
> These results support the effectiveness of our length-control strategy: overly long outputs risk truncation, while overly short outputs hinder exploration.
>
> [1]  An Yang, et al. "Qwen2.5-Math Technical Report: Toward Mathematical Expert Model via Self-Improvement"

---

### Official Review · Reviewer_CwVY · 2025-11-03

**Soundness:** 2
**Presentation:** 3
**Contribution:** 2
**Rating:** 4
**Confidence:** 4

**Summary:**

This work studies training LLMs with surrogate rewards on reasoning tasks —a format check and a piecewise length reward— arguing these signals can stand in for ground-truth correctness. The method targets the early regime where “format learning” dominates, and claims that adding a length prior prevents saturation and nudges the model toward more effective reasoning. On Qwen2.5-Math and OctoThinker (based on Llama3.1-8B), the authors report that format-only quickly matches correctness-based GRPO in the first ~15 steps but plateaus, while format+length (their main method) achieves competitive or occasionally better accuracy than correctness-based GRPO. They also include experiments on uncontaminated 2025 test sets and analyze response-length dynamics and reflective-word usage.

**Strengths:**

- The paper is generally well-written and experimental details are clear.
- Results are across multiple datasets with care taken regarding decontamination and across model sizes and families.
- Authors provide additional qualitative analysis (response-length trends, reflective-word frequencies, etc) and ablations (eg. RL vs SFT for format learning) that align with their findings.

**Weaknesses:**

- My main concern is the novelty of the work; the author’s central claim– that simple structure-based signals (format, length) can substitute correctness in GRPO– are preceded by several prior works which study RLVR without external supervision rewards [1,2] as well as study whether RL is eliciting capabilities beyond what can be achieved by the base model [3,4,5], which are not cited in this work (the above examples I gave is not exhaustive– the authors do cite [6]).
- In particular, [2] shows very similar results in the random reward setting, and it seems that the results found by this work are by virtue of the choice of Qwen 2.5 Math base model and bias introduced by things like clipping with the GRPO algorithm - indeed, using the correctness reward is better for OctoThinker as we see in Table 2.
- Since the efficacy of the methods in this work seem to be highly model-dependent (as the authors themselves note: “If the base model is not powerful, we expect this approach to yield limited gains.”), I think the work would be much stronger if the authors did further ablations regarding the design of their length-based reward, verified against a correctness + length baseline, and when one can expect their surrogate signal to capture most of the gains from a correctness-based reward given a choice of base model (for instance, relating pass @ 64 performance or ability to benefit from additional response tokens). Please see ‘Questions’ for more specific details in this regard.

Minor:
- Some areas in the manuscript need `\citep` (eg. citations in the Introduction).
- Wording in sentence from lines 214-215
- Line 375: thes -> these
- Line 729: A.5 Introducion -> Introduction

[1] Agarwal, Shivam, et al. "The unreasonable effectiveness of entropy minimization in llm reasoning." arXiv preprint arXiv:2505.15134 (2025).
[2] Shao, Rulin, et al. "Spurious rewards: Rethinking training signals in rlvr." arXiv preprint arXiv:2506.10947 (2025).
[3] Liu, Mingjie, et al. "Prorl: Prolonged reinforcement learning expands reasoning boundaries in large language models." arXiv preprint arXiv:2505.24864 (2025).
[4] Wu, Fang, et al. "The Invisible Leash: Why RLVR May or May Not Escape Its Origin." arXiv preprint arXiv:2507.14843 (2025).
[5] Zhao, Rosie, et al. "Echo chamber: Rl post-training amplifies behaviors learned in pretraining." arXiv preprint arXiv:2504.07912 (2025).
[6] Yue, Yang, et al. "Does reinforcement learning really incentivize reasoning capacity in llms beyond the base model?." arXiv preprint arXiv:2504.13837 (2025).

**Questions:**

- Did the authors observe any instances where their format+length reward did not capture most of the performance gains from a correctness-based reward? Negative results in their setting are still informative.
- Could the authors include a correctness + length-based reward baseline (weighted average, or add length reward only on correct responses)?
- In Shao et al. they observe that their performance gains from random rewards may be due to bias in the GRPO formulation, particularly clipping focuses the model on its existing reasoning pattern distribution. If the authors remove clipping effects (either by removing clipping entirely, or making the rollout policy identical to the updating policy) do they observe the same effects?
- How sensitive are the authors findings to their choice of p in their length reward?
- What are the pass @ 64 performance metrics for the other models (I only saw numbers for Qwen 2.5 7B in Table 4, I apologize if I missed the others)? Does this correlate with OctoThinker’s comparatively worse relative gains from the format+length reward? More broadly, what are the authors’ thoughts on a necessary quantitative condition for the base model to fulfill for their proposed method to work effectively?

---

> ### Author Response · Authors · 2025-11-25
> **The 1/2 of the reply**
>
> We sincerely thank the reviewers for their careful and accurate comments. In Appendix A.3 of the paper, we have already cited several works on RLVR without external supervision. For the works you mentioned, we will add the corresponding citations in the final version.
>
> ---
>
> ## **Q1**
>
> Yes. As stated in the paper, our method shows limited improvement on base models without dedicated mathematical pre-training.
>
> ---
>
> ## **Q2  Experiments on Correctness + Format-Length**
>
> Motivated by the availability of correctness signals, we designed a method to combine correctness rewards with the existing format and length rewards. The scoring rule is:
>
> ```python
> if correctness_reward > 0:
>     score = correctness_reward + (format_reward + length_reward) * 0.2
> else:
>     if format_reward > 0:
>         score = (format_reward + length_reward) * 0.1
>     else:
>         score = min(0.0, length_reward)
> ```
>
> The idea is:
>
> * when the model answers correctly, we upweight the correctness reward;
> * when the answer is incorrect but the format is valid, we provide a small length-based reward;
> * when both correctness and format are invalid, we penalize length (especially if overlong).
>
> The results on **Qwen2.5-Math-7B** are:
>
> | Method                     | AIME2024 | MATH500 | AMC2023 | Avg.  |
> | ------------------------- | -------- | ------- | ------- | ---- |
> | Format-Length             | 33.3     | 33.3    | 60.2    | 56.8 |
> | Correctness+Format-Length | 26.7     | 78.2    | 61.5    | 55.4 |
>
> We observe that introducing correctness rewards does **not** provide additional gains, whereas the **format+length** surrogate signal already captures most of the benefits.
>
> ---
>
> ## **Q3  On-policy Training and the Effect of Clipping**
>
> All of our experiments are conducted using the **VERL** framework with:
>
> ```
> train_batch_size = ppo_mini_batch_size = 128
> ```
>
> which is theoretically **on-policy** and eliminates clipping-induced bias.
>
> To further remove unintended clipping effects caused by numerical discrepancies between **vLLM rollout** and **actor model forward (FSDP)** [1], we modified VERL to **disable clipping entirely**. Results on Qwen2.5-Math-7B:
>
> | Method                        | AIME2024 | MATH500 | AMC2023 | Avg.  |
> | ---------------------------- | -------- | ------- | ------- | ---- |
> | Format-Length (on-policy)    | 33.3     | 76.8    | 60.2    | 56.8 |
> | Format-Length (disable_clip) | 26.7     | 76.8    | 61.5    | 55.0 |
>
>
> Overall, our method still yields **significant gains even when clipping is completely removed**, demonstrating that the effectiveness of our approach **does not rely on clipping-induced bias**.
>
> ---
>
> ## **Q4  Ablation on the Parameter *p***
>
> In the paper, the default value is *p = 0.5*. We additionally tested *p = 0.4, 0.6, 0.8*:
>
> | Method   | AIME2024 | MATH500 | AMC2023 | Avg.      |
> | ------- | -------- | ------- | ------- | -------- |
> | p = 0.5 | 33.3     | 76.8    | 60.2    | 56.8     |
> | p = 0.4 | 33.3     | 77.2    | 59.0    | 56.5     |
> | p = 0.6 | 30.0     | 75.8    | 60.2    | 55.4     |
> | p = 0.8 | 33.3     | 78.6    | 60.2    | **57.4** |
>
> All three variants perform similarly to *p = 0.5*, with *p = 0.8* being slightly better. This suggests the method is **not sensitive** to the choice of *p*.
>
> We also tested extreme values:
>
> * **p = 1.0** (no penalty for long responses):
>   similar to the *LinearLength* setting discussed in Sec. 4.3—response length quickly approaches the context limit, causing truncation.
> * **p = 0.1**:
>   response lengths shrink throughout training, limiting exploration and eventually causing a decline after an initial spike.
>
> These results support the effectiveness of our length-control strategy: overly long outputs risk truncation, while overly short outputs hinder exploration.
>
> ---

---

> ### Author Response · Authors · 2025-11-25
> **The 2/2 of the reply**
>
> ## **Q5  Additional pass@64 Results**
>
> | Model                               | AIME2024 | MATH500 | AMC2023 |
> | ----------------------------------- | -------- | ------- | ------- |
> | Llama3.1-8B (OctoThinker-8B-Hybrid) | 26.7     | 88.8    | 77.1    |
> | Qwen-Math-1.5B                      | 40.0     | 91.4    | 84.3    |
>
> Interestingly, **OctoThinker-8B** performs significantly worse than the much smaller **Qwen-Math-1.5B** on pass@64.
> Up to now, we have not yet identified a fully quantitative method for this evaluation (though **pass@64** may be a promising direction). From a qualitative perspective, our method is most effective on base models that have undergone dedicated **mathematics pretraining**. On such models, our alternative-signal approach achieves **over 90% correctness substitution**, as shown in the *All Models Performance Gain* section of Appendix below.
>
> Moreover, our motivation is to show that reinforcement learning in post-training primarily **activates knowledge already learned during pretraining**, rather than introducing new knowledge. The fact that our method—**which uses no labeled data**—works well on math-pretrained models directly supports this claim. This further suggests a shift in how resources might be allocated across the LLM training pipeline: **investing more heavily in pretraining**, where real “intelligence injection” occurs.
> It's worth mentioning that Gemini3 has recently achieved great success, and the co-leader of the Gemini project confirmed that the scaling principles of pre-training remain effective, indicating that pre-training has far from reached its limits.
>
> ---
> [1] Yao, Feng, et al.  "Your Efficient RL Framework Secretly Brings You Off-Policy RL Training"
>
> # **Appendix**
>
> ## **Additional Models**
>
> We evaluated additional architectures. On **deepseek-math-7b-base**, our method **surpasses** the baseline; on **Mathstral-7B-v0.1**, Format-Length reaches **96%** of the correctness-based method. This further demonstrates the **robustness and generality** of our technique when applied to strong math pretrained base models.
>
>
> | Method                 | AIME2024 | MATH500  | AMC2023  | Avg.      |
> | --------------------- | -------- | -------- | -------- | -------- |
> | Deepseek-Math-7B-Base | 0.0      | 1.0      | 0.0      | 0.3      |
> | Correctness           | 0.0      | 22.4     | **16.9** | 13.1     |
> | Format-Only           | 0.0      | **27.4** | 12.1     | 13.2     |
> | Format-Length         | **3.3**  | 25.8     | 12.1     | **13.7** |
>
>
> | Method             | AIME2024 | MATH500  | AMC2023  | Avg.      |
> | ----------------- | -------- | -------- | -------- | -------- |
> | Mistral-7B(Mathstral-7B-v0.1)  | 0.0      | 51.4     | 19.3     | 23.6     |
> | Correctness       | **6.7**  | 61.8     | 30.1     | **32.9** |
> | Format-Only       | 3.3      | 52.0     | 27.7     | 27.7     |
> | Format-Length     | 0.0      | **62.8** | **32.5** | 31.8     |
>
> ---
> ## **All Models Performance Gain**
> Across *all* model families, our Format+Length reward achieves **>90%** of correctness-based RL performance.
>
> | Model                        | Init | Correctness   | Format-Length | Format-Length / Correctness (%) |
> | ---------------------------- | ---- | ------------- | ------------- | ------------- |
> | Qwen-Math-1.5B               | 27.1 | 43.1 (↑59%)   | 43.5 (↑60%)   | 100.9%        |
> | Qwen-Math-7B                 | 36.6 | 53.0 (↑45%)   | 56.8 (↑55%)   | 107.2%        |
> | Qwen-Math-72B                | 56.2 | 64.5 (↑15%)   | 62.7 (↑12%)   | 97.2%         |
> | Llama3.1-8B (OctoThinker-8B) | 18.7 | 38.7 (↑107%)  | 35.7 (↑91%)   | 92.2%         |
> | Deepseek-Math-7B-Base        | 0.3  | 13.1 (↑4267%) | 13.7 (↑4467%) | 104.6%        |
> | Mistral-7B(Mathstral-7B-v0.1)  | 23.6 | 32.9 (↑39%)   | 31.8 (↑35%)   | 96.7%         |
>
> (*The scores in the table are the average of the three evaluation sets (AIME2024, MATH500, AMC2023).Values in parentheses indicate improvement relative to the initial model.*)
>
> ---
>
> ## **Evaluation on OOD Datasets**
>
> Since our training data contains only math tasks, these datasets are out of distribution ( **OOD** ). Yet, our method still outperforms baselines, especially on the hardest code generation benchmark **CruxEval (Output)**, strongly demonstrating the **generalization capability** of our approach.
>
> | Method                | GPQADiamond | HumanEvalPlus | MBPPPlus | CruxEval(Output) | Avg.      |
> | --------------------- | ----------- | ------------- | -------- | ---------------- | -------- |
> | Qwen-Math-7B       | 24.2        | 52.4          | 40.5     | 27.5             | 36.2     |
> | Correctness           | 25.8        | 51.8          | **52.6** | 20.9             | 37.8     |
> | Format-Only           | 25.3        | **62.2**      | 50.5     | 32.8             | 42.7     |
> | Format-Length (p=0.8) | **29.8**    | 52.4          | 50.0     | **57.6**         | **47.5** |

---

### Author Response · Authors · 2025-12-03
**The 3/3 to Area Chair**

### 3.4 Out-of-distribution generalization

Reviewers XzHt and 6Aps wondered if format/length-only supervision would fail on non-math reasoning tasks.

Therefore, we evaluated on a **completely OOD code and inference benchmark** (training data consisted only of mathematical data):

* GPQA-Diamond
* HumanEvalPlus / MBPPPlus / CruxEval (Output)

On **Qwen2.5-Math-7B**, Format+Length outperforms both the base model and correctness-RL, especially on difficult OOD tasks like CruxEval:

| Method                | GPQAD    | HumanEvalPlus | MBPPPlus | CruxEval(Output) | Avg.     |
| --------------------- | -------- | ------------- | -------- | ------------- | -------- |
| Qwen2.5-Math-7B       | 24.2     | 52.4          | 40.5     | 27.5          | 36.2     |
| Correctness           | 25.8     | 51.8          | **52.6** | 20.9          | 37.8     |
| Format-Only           | 25.3     | **62.2**      | 50.5     | 32.8          | 42.7     |
| Format-Length (p=0.8) | **29.8** | 52.4          | 50.0     | **57.6**      | **47.5** |

**Conclusion:** Even without correctness supervision, Format+Length yields meaningful gains on unseen reasoning tasks, alleviating concerns that it was "only mathematically effective".

---

### 3.5 Sensitivity to the length parameter *p*

Reviewers CwVY and XzHt asked about hyperparameter sensitivity.

We swept p = 0.4, 0.6, 0.8 on Qwen2.5-Math-7B:

| Model                 | AIME2024 | MATH500 | AMC2023 | Avg.  |
|-----------------------|----------|---------|---------|------|
|  p=0.5 (in paper) | 33.3     | 76.8    | 60.2    | 56.8 |
|  p=0.4   | 33.3     | 77.2    | 59.0    | 56.5 |
|  p=0.6   | 30.0     | 75.8    | 60.2    | 55.4 |
|  p=0.8   | 33.3     | 78.6    | 60.2    | **57.4**|

* Minimal performance differences , p = 0.8 slightly better;

* Extreme values: p = 1.0 (infinite scaling) leads to length explosion until truncation; p = 0.1 causes premature contraction and insufficient exploration.

 **the method is insensitive to p and has a very low parameter tuning burden**

---

### 3.6 Risks of repetition or mechanical lengthening

Reviewer 6Aps worried that length rewards might induce repetition or mechanical expansion.

Our analysis (Appendix A.4) shows:

* **no increase** in repeated substring ratios;
* no pathological length inflation due to the “rise-then-fall” length curve;
* OOD coding results further validate absence of mechanical repetition.

Thus the length reward does **not** induce known RL failure modes.

---

### 3.7 Generality of the format reward and online SFT

Reviewer XzHt asked about the role of online SFT.

Online SFT here means training on model-generated data filtered by the format reward. This serves as an ablation: **online SFT and GRPO with format reward behave similarly**, reinforcing that format learning is effective even outside the GRPO algorithmic setup.

---

## 4. Clarifying the Main Message

Reviewer XzHt asked whether the overarching message is clear.

We emphasize two points:

1. By using surrogate signals, the effect can approximate or even surpass that of real labels in some cases. Format+Length offers an extremely effective, easy-to-use alternative when labels are expensive, scarce, or risky (e.g., leakage concerns).

2. More fundamentally:

   > Unsupervised RL achieved similar results to using labeled data, which indirectly proves that pre-training is the stage where true intelligence is injected. The similar pass@64 scores of the model before and after reinforcement proves that RL only optimizes the way the model retrieves and organizes existing knowledge.


---

## 5. Conclusion: Why This Paper Should Be Accepted

This work has the following strengths:

1. **Simplicity, reproducibility, and high engineering value**
   A minimalist, label-free RL approach that is directly useful to both industry and academia.

2. **Extensive, rigorous experiments**
   Covering multiple model families (Qwen, Llama, Deepseek, Mistral, Qwen3), many datasets (including decontaminated and OOD task),  qualitative analysis (reflective-word frequencies, etc) and numerous ablations (clipping, correctness+Format+Length, p sweep, online SFT).

3. **New insights into RL mechanisms**
   Concrete, quantitative evidence for the “RL activates pretrained knowledge” hypothesis, providing guidance for future label-efficient RL design.

4. **All key reviewer concerns have been thoroughly addressed**

   * Novelty → clarified and supported by new experiments
   * Generalization → validated across many architectures and OOD tasks
   * Clipping / hyperparameters → ruled out as confounders
   * Length/repetition risks → empirically shown not to occur

For these reasons, we sincerely hope that you will consider **accepting this submission**. It offers both **a practical label-free RL technique** and **a systematic, insightful empirical study** of RL’s mechanism in LLM reasoning—contributions that we believe will have lasting impact on the community.

Respectfully,
The Authors

---

### Author Response · Authors · 2025-12-03
**The 2/3 to Area Chair**

### 3.2 Clipping bias concerns

Reviewer CwVY asked whether our findings simply reflect GRPO clipping bias.

We addressed this in two ways:

1. All paper experiments use **strict on-policy training** (train_batch_size = ppo_mini_batch_size = 128), eliminating off-policy clipping bias.
2. We conducted **fully unclipped training** in VERL, removing any hidden clipping effects.

Results (Qwen2.5-Math-7B):

| Method                       | AIME2024 | MATH500 | AMC2023 | Avg. |
| ---------------------------- | -------- | ------- | ------- | ---- |
| Format-Length (on-policy)    | 33.3     | 76.8    | 60.2    | 56.8 |
| Format-Length (disable_clip) | 26.7     | 76.8    | 61.5    | 55.0 |

**Conclusion:** Surrogate rewards remain beneficial even without clipping.
Thus, our effect is **not** due to clipping bias.

---

### 3.3 Multi-model, cross-architecture validation

Reviewers 6Aps and CwVY questioned whether the method generalizes beyond Qwen models.

We added extensive experiments on **other pretrained models with other architectures** , including:

* Deepseek-Math-7B-Base
| Model                   | AIME2024 | MATH500 | AMC2023 | Avg.  |
|-------------------------|----------|---------|---------|------|
| Deepseek-Math-7B-Base   | 0.0      | 1.0     | 0.0     | 0.3  |
| Correctness             | 0.0      | 22.4    | **16.9** | 13.1 |
| Format-Only             | 0.0      | **27.4** | 12.1    | 13.2 |
| Format-Length           | **3.3**  | 25.8    | 12.1    | **13.7** |

* Mistral-7B (Mathstral-v0.1)

| Model               | AIME2024 | MATH500 | AMC2023 | Avg. |
|---------------------|----------|---------|---------|------|
| Mistral-7B (Mathstral-7B-v0.1)| 0.0 | 51.4 | 19.3  | 23.6 |
| Correctness         | **6.7**  | 61.8    | 30.1    |**32.9**|
| Format-Only         | 3.3      | 52.0    | 27.7    | 27.7 |
| Format-Length       | 0.0      | **62.8**| **32.5**| 31.8 |

* Qwen3-30B-A3B-Base (with decontaminated datasets)

| Model                  | AIME2025 | MINERVA | LIVE_MATH_BENCH2025  | AIME2024 | AMC2023 | MATH500 | Avg.  |
|------------------------|----------|---------|----------------------|----------|---------|---------|------|
| Qwen3-30B-A3B-Base     | 10.0     | 14.3    | 9.1                  | 20.0     | 41.0    | 70.0    | 27.4 |
| Correctness            |**13.3**  |**24.6** | 13.1                 |**26.7**  |**60.2** | 80.4    |**36.4**|
| Format-Only            |**13.3**  | 23.9    |**14.1**              | 20.0     | 56.6    | 80.0    | 34.7 |
| Format-Length          |**13.3**  | 23.9    | 12.1                 |**26.7**  | 53.0    |**81.2** | 35.0 |



* **Summary across all models**:

| Model                               | Init     | Correctness    | Format-Length   | Format-Length / Correctness (%) |
|-------------------------------------|----------|----------------|-----------------|----------------|
| Qwen-Math-1.5B                      | 27.1     | 43.1 (↑59%)    | 43.5 (↑60%)     |**100.9%**       |
| Qwen-Math-7B                        | 36.6     | 53.0 (↑45%)    | 56.8 (↑55%)     |**107.2%**       |
| Qwen-Math-72B                       | 56.2     | 64.5 (↑15%)    | 62.7 (↑12%)     | 97.2%          |
| Llama3.1-8B (OctoThinker-8B-Hybrid) | 18.7     | 38.7 (↑107%)   | 35.7 (↑91%)     | 92.2%          |
| Deepseek-Math-7B-Base               | 0.3      | 13.1 (↑4267%)  | 13.7 (↑4467%)   |**104.6%**       |
| Mistral-7B(Mathstral-7B-v0.1)       | 23.6     | 32.9 (↑39%)    | 31.8 (↑35%)     | 96.7%          |
| Qwen3-30B-A3B-Base 		          | 27.4     | 36.4 (↑33%)    | 35.0 (↑28%)     | 96.2%          |

(*Values in parentheses indicate improvement relative to the initial model.*)

Across all model families:

* Format+Length reaches **>90%** of correctness RL.
* For some models (such as Deepseek-Math-7B-Base, Qwen-Math-1.5B/7B),  it **exceeds** correctness RL.
* Llama/Mistral models achieve **92–97%** correctness RL under Format+Length.

**Conclusion:** The method is **robust and generalizes across architectures**, not a Qwen-specific phenomenon.

---

### Author Response · Authors · 2025-12-03
**The 1/3 to Area Chair**

Dear Area Chair,

Thank you for your careful evaluation of our work, and for coordinating the detailed and constructive reviews. Below, we summarize the **core contributions**, **value to the community**, and the **additional experiments and clarifications** we performed in response to reviewer feedback. We also explain why we believe this submission merits acceptance.

---

## 1. Scope and Core Contributions

This work focuses on a simple but powerful idea: **in mathematical reasoning tasks, RL post-training can be effectively driven by two extremely simple surrogate signals—format + length—without using any labeled data, while still approximating or even surpassing correctness-based RL.**

Our main contributions are threefold:

1. **A remarkably simple and fully reproducible surrogate reward design**

   * **Format reward:** checks only whether the model outputs a valid final answer wrapped in `\boxed{}`.
   * **Length reward:** a piecewise “rise-then-fall” function that encourages moderately long chain-of-thought reasoning.

   These signals require **no ground-truth labels** and are **trivially reproducible**, a point acknowledged by all reviewers.

2. **The first systematic characterization of a two-stage RL phenomenon: early “format learning” + later “length-driven refinement”**

   * During the **first ~15 GRPO update steps**, format-only signals yield **most of the accuracy gain**.
   * Adding length rewards later **further boosts performance** by discouraging both overly short and overly long outputs.

   This structured, quantitative analysis of RL training dynamics fills a gap in the current RLVR literature, which lacks such systematic empirical characterization.

3. **Propose and validate a thought-provoking viewpoint on the mechanism of RL**

* Through a series of "unlabeled/structured reward" experiments:

> RL in inference tasks primarily **activates existing knowledge from pre-training**, rather than "learning" new mathematical knowledge.

* On various pre-trained base models, our Format+Length reward achieves **>90%** correctness-RL performance (and even surpasses it in some cases), without requiring labels. This fact provides **new empirical evidence** and a **more refined procedural perspective** for the ongoing community discussion on whether "RL truly expands the capabilities of base models."


---

## 2. Relation to RLVR Work and Novelty

Reviewer CwVY noted similarities to existing label-free RLVR work (e.g., entropy minimization, spurious rewards) and expressed concerns about novelty.

We have **clarified these distinctions in the paper and appendix**, and further addressed them in rebuttal:


1. **Different problem setting**

   Prior work largely explores:
   * how random/spurious rewards can still yield improvements;
   * whether RL truly expands the model’s capability frontier.

   In contrast, we provide:
   * **a constructive, interpretable surrogate reward (format + length)**;
   * **a systematic study** of when and why it can **replace** correctness rewards and evidence supporting a more refined view of RL’s mechanism.

2. **More constructive, less diagnostic**

   Prior works tend to present *negative/diagnostic* cases (e.g., random reward gains, clipping issues).

   Our work provides a **practical, label-free RL strategy** validated across many models and datasets, highlighting its *operational value*.

3. **New, systematic empirical evidence**

   We added extensive experiments during rebuttal (summarized in Section 3 below), further strengthening our independent contribution.

(*Our work and other RLVR-unsupervised projects occurred around the same period, as evidenced by release dates in the community.*)

Therefore, **our work delivers a minimal, reproducible surrogate reward design, and a systematic empirical foundation for understanding “RL as activation of pretrained knowledge” within the label-free RLVR paradigm.**

---

## 3. Additional Experiments: Addressing Reviewer Concerns

### 3.1 Correctness + Format-Length baseline

Reviewers CwVY and XzHt asked whether combining correctness with Format+Length could outperform either alone.

We designed a simple combined reward:

```python
if correctness_reward > 0:
    score = correctness_reward + (format_reward + length_reward) * 0.2
else:
    if format_reward > 0:
        score = (format_reward + length_reward) * 0.1
    else:
        score = min(0.0, length_reward)
````

Results on **Qwen2.5-Math-7B**:

| Method                    | AIME2024 | MATH500 | AMC2023 | Avg. |
| ------------------------- | -------- | ------- | ------- | ---- |
| Format-Length             | 33.3     | 76.8    | 60.2    | 56.8 |
| Correctness+Format-Length | 26.7     | 78.2    | 61.5    | 55.4 |

**Conclusion:** Adding correctness provides **no additional benefit** .
Thus:

> **Format+Length already captures the primary gain of correctness-RL, and serves as a strong substitute when labels are costly or unavailable.**

---

### Meta-Review · Area_Chair_Rk3A · 2026-01-02

**Summary:**

This paper studies RL for mathematical reasoning using surrogate reward signals based on output format and response length, instead of ground-truth correctness. The authors show that early GRPO training is dominated by format learning, and that adding a length-based reward helps recover much of the performance of correctness-based RL on several math benchmarks. The paper further argues that RL primarily activates knowledge already present in pre-trained models rather than introducing new knowledge. Across reviews, there was broad agreement that the paper is clearly written, carefully executed, and empirically thorough. Reviewers appreciated the simplicity and reproducibility of the reward design, as well as the detailed analysis of training dynamics. However, reviewers consistently raised concerns about limited novelty relative to recent RLVR and "spurious reward" literature, the dependence on strong math-pretrained base models, and the incremental nature of the conceptual contribution. While the rebuttal addressed many technical questions and added substantial experiments, these concerns ultimately remained.

**Reviewer Concerns:**

Overall, the rebuttal successfully addressed most *technical questions*, but did not make a compelling case on novelty and significance. CwVY raised concerns about novelty relative to prior work on label-free RL and spurious rewards, possible confounding effects from GRPO clipping, and whether correctness+length would outperform either alone. The authors addressed these points with additional experiments (including on-policy and no-clipping variants, correctness+length baselines, and sensitivity analyses). These additions clarify that the reported effects are not artifacts of clipping and that the surrogate rewards capture much of the correctness signal. However, the reviewer's underlying concern about limited conceptual novelty relative to recent RLVR work remains. XzHt questioned the clarity of the paper's main message, the statistical significance of small gains over correctness-based RL, and whether format/length supervision would generalize to harder or longer reasoning settings. The rebuttal improved clarity around the intended scope (label-scarce settings) and addressed hyperparameter sensitivity, but did not substantially strengthen claims about broader generalization or impact beyond the studied benchmarks. 6Aps focused on generalization beyond Qwen and risks of mechanical lengthening or repetition. The authors added experiments on additional model families and analyses showing no increase in repetition, which largely address these technical concerns. However, the results reinforce that the method is most effective when the base model already has strong latent mathematical capability, limiting the generality of the contribution.

**Reviewer Scores:**

- For CwVY the rebuttal addressed technical questions and removes some uncertainty, but does not resolve the core novelty concern. Score would likely stay at a 4.
- For XzHt the rebuttal clarifies intent and adds experiments, but the reviewer's concerns about impact, statistical significance, and generality would likely persist.
- For 6Aps the added experiments and repetition analysis might nudge this to a 5 for them.

---

### Decision · Program_Chairs · 2026-01-26

Reject